# Assessing Livestock Grazing Distribution in Communal Rangelands of the Eastern Cape, South Africa: Towards Monitoring Livestock Movements in Rangelands

Bukho Gusha [1,*], Anthony R. Palmer [1] and Thantaswa C. Zondani [2]

1   Institute for Water Research, Rhodes University, Grahamstown 6140, South Africa
2   Agricultural Research Council, Animal Production, Grahamstown 6140, South Africa
*   Correspondence: b.gusha@ru.ac.za

**Abstract:** In the past, rangelands were managed in a semi-nomadic manner, where pastoralists would distribute livestock to different parts of the rangeland depending on the availability of forage. However, understanding how livestock use rangelands has not been a subject of many studies as the devices to monitor livestock were not available. The objective of this study was to assess livestock grazing distribution in communal rangeland of the Eastern Cape in South Africa in order to improve livestock grazing. The study used Global Positioning Systems (GPS) that were placed on six animals including three sheep and three cattle, selected randomly from participating households. The GPS collars recorded the geographic position of the areas where animals were grazing, at five-minute intervals during the wet and dry season. Grass species composition was surveyed using line transects on areas where livestock grazing occurred. The study further identified three production domains, which were separated by bound polygons on Google Earth Pro to extract MODIS EVI where livestock grazing occurred. Livestock grazing distribution was analysed using T-LoCoH installed in R. The results revealed that both sheep and cattle spent most of their time grazing in areas associated with human features. The dominant grass species were Hyperrenia hirta and Eragrostis plana, suggesting a negative impact of the current livestock grazing distribution. Possible explanations of the current grazing distribution might be that these areas are close to homesteads and provide grazing lawns that contain a high nitrogen content. This study will help in providing an informed basis for the development of South African communal rangeland policies for effective livestock management.

**Keywords:** cattle; communal rangelands; Eastern Cape; grazing lawns; livestock

## 1. Introduction

In many parts of Africa, rangelands are an important support system for the livelihoods of millions of people, livestock production, and biodiversity conservation [1]. However, because of climatic factors and human impact on the environment, many rangelands are considered severely degraded, impacting negatively on the people who rely on land and livestock production [2,3]. Rangeland degradation is chiefly related to poor grazing management, which is often reported as a major contributor to reduced rangeland stability [4]. In areas where rangeland degradation is severe, communal rangelands are the most degraded ecosystems and sustainable grazing management is important to improve the value of the rangelands in terms of their socio-economic and ecological systems [4,5]. In South Africa, communal rangelands make up only 13% of agricultural land on account of colonialism, under which the indigenous people were restricted to small areas and land rights were delineated along racial lines through the 1913 Native Land Act [6]. These communal rangelands were administered by local chiefs or village headmen who determined who could own livestock on the common rangeland or 'amadlelo', how many livestock were permitted per household, and where the livestock should graze at different times of the year [7]. With the implementation of the restrictive legislation after 1913, and the

"Betterment Planning" following the recommendations of the Tomlinson Commission in 1954, traditional livestock management governance became fractured. Villages could no longer practice the principles embedded in a herding culture, as camps (or paddocks), gates, and fences replaced the decisions of herders [7]. Under the herding system, livestock roamed freely and were guided to graze on the most productive areas on the communal grazing [8]. However, at the household level, farmers seldom have the resources to employ a herder to guide livestock to different grazing areas throughout the year. The loss of herding skills, as well as mandatory school attendance laws [9], further eroded the role of herders in determining where livestock should graze in different seasons.

Against this drop back, many parts of the rural Transkei and other dry land regions have been subjected to ploughing, followed by large-scale abandonment, leading to substantial changes in the species composition of the rangeland, which are now dominated by the more robust, less palatable grasses such as *Eragrostis plana*, *Hyperrenia hirta*, and *Sporobolus africanus* [10]. These grass species are an indication of a rangeland that is in poor condition due to disturbances such as overgrazing. This makes it important to understand livestock grazing distribution in communal rangelands as it affects species composition, and when not well monitored, leads to accelerated soil erosion. This is equally because livestock make use of a patchy mosaic of available forage in time and space [11], in order to maximize intake. An understanding of livestock grazing distribution coupled with knowledge of the factors limiting the ability of the farmers to do this effectively, will enable graziers to think about ways to maximize forage utilization and potentially improve livestock production. Living and non-living factors influence livestock preference for some sites over others, as well as animal behavior [6]. According to [12], in a free ranging situation, livestock selectivity is driven by several factors such as plant type (grass, forbs, and shrubs), forage quality, quantity and palatability, plant species, shade and shelter, human activities, insects, pests, soil, weather, topography, and water. However, in a controlled livestock production system (rotational grazing), graziers are able to control some of these factors in the activity of livestock, through timing and managing the duration of stay in a paddock.

In order to improve the understanding of how livestock use rangelands in a continuous grazing situation in the communal rangelands of the Eastern Cape, this study used Global Positioning Systems (GPS) to identify patterns of rangeland use where interventions would be most beneficial to improving livestock production and rangeland management. The collars were placed on selected cattle and sheep under a continuous grazing system to analyze the grazing distribution of the two animal types. This was necessary to help in providing an informed basis for the development of South African communal rangeland policies for effective livestock management. An assessment of the areas that provide grazing for livestock and particularly those that provide key resources in maintaining livestock during the wet and dry seasons and when these areas are utilized was determined. These areas such as unimproved grasslands, cultivated lands, and areas around homesteads were determined following [13]. Grass species composition was determined to record the most dominant grass species in the frequently grazed areas.

## 2. Materials and Methods

### 2.1. Site Description

The study site is in the northern part of Eastern Cape Province, South Africa, in the former Transkei homelands near the town of Cala, and the village of Mgwalana. It is in the quaternary river catchment T12A (centered around 31°31′25 S; 27°45′27 E) that is administered under a communal tenure arrangement (Figure 1). Livestock herds are mostly dominated by sheep and cattle, with some herds including a few goats [14]. The study site comprises vegetation from the Drakensberg foothills moist grassland [15], which can be found across a broad arc of Drakensberg mountains and their surroundings. The topography ranges from steep mountainous escarpments to moderately rolling grassland incised by river gorges. Mudstones and sandstones of the Molteno formation and Tarkastad, as well as intrusive dolerites of the Jurassic Age, dominate the geology in the study site.

Well-drained soils on the sedimentary parent material dominate, with clay content ranging from 15–55% and a depth of more than 800 mm, including soil forms such as Hutton, Griffin, Oakdale, and Clovelly. On the dolerites, soil forms include Mispah rock complex, Shortlands, Balmoral, and Vimy. The study site receives 654 mm (2000–2017) of mean annual precipitation during the summer rainfall period [15]. There are 26 frost days, indicative of a sub-montane form of a warm temperate climate, and the mean annual temperature is 14.6 °C. The study site contains important taxa of graminoids and geophytic herbs [15].

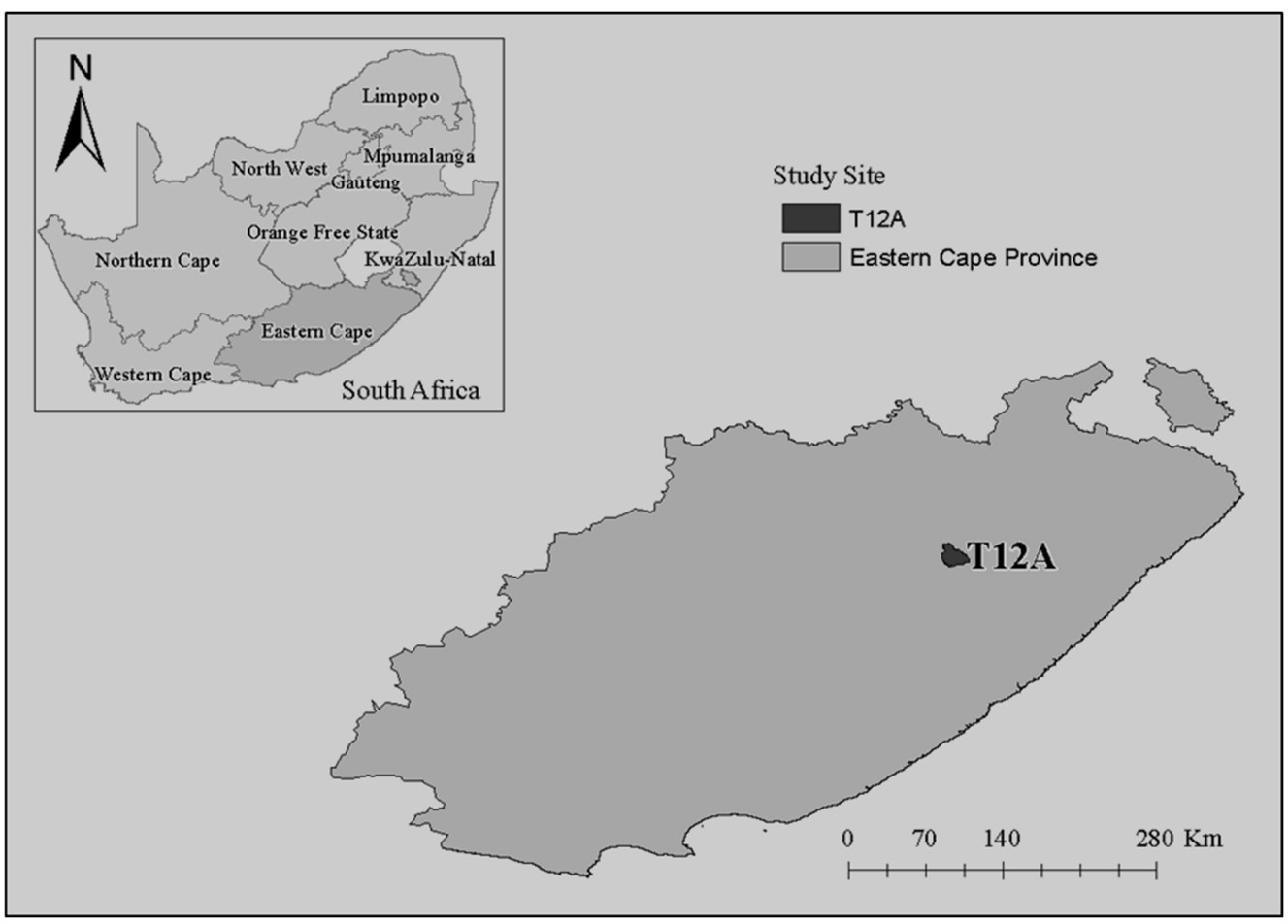

**Figure 1.** Map of the study site where animal grazing distribution occurred in the village of Mgwalana (T12A).

### 2.2. Experimental Livestock and Mounting of GPS Collars

Two data collection expeditions were conducted during the wet (November 2016 to February 2017) and dry season (July 2017 to September 2017). Three local zebu-type cattle (Bos indicus) and three non-descript local sheep were selected as focal animals from herds of collaborating homesteads. Each animal was fitted with a CatLog GPS tracking device inserted into a secure, waterproof pouch sealed with silicon to prevent water damage and attached to a robust neck collar belt, individually adjusted to their neck size. The devices were inserted in collar belts that were placed on animals from the 22 November 2016 to 18 February 2017 to record grazing distribution for the wet season. From the 4 July 2017 to 5 September 2017, the same collar belts with devices were placed on animals to record grazing distribution for the dry season. After retrieving the collars, the position data were extracted from the GPS trackers using the CatLog Software. Out of the six animals collared (three sheep and three cattle), only one animal species was used in each season to represent an isopleth for the proportion of the areas used by the focal animal.



### 2.3. Data Preparation, Cleaning, and Generation of Google Earth Images

Data from the GPS trackers were downloaded using the CatLog Data Centre and stored in a comma delimited ASCII text format (*.csv). Data files were renamed to follow a unique numerical convention starting with ARC01, for the first collar activated in summer, and ARC01W for the same collar activated in winter. All animals were weighed at the time of collar attachment, and the data were stored in a database linked to the collar number. The timestamp on data from the CatLog GPS receivers in Greenwich Mean Time (GMT) for use in South Africa, was changed. Using the R packages lubricate and plyr, data from each collar were converted into Central African Time (CAT) and day/time segments. The day/time segments for the summer data were 6 am to 6 pm and described as day and night. The location data from the CatLog GPS receivers were in a geographic projection (latitude; longitude), but as this was a pseudo-projection, it was not possible to calculate area or distance, and conversion to recognized projection was required. In this study, the universal transverse Mercator (UTM) was selected as it was required for further analysis in T-LoCoH [16].

In order to reduce the incidence of collar failure, all GPS collars were activated in the laboratory prior to placing them inside the waterproof containers. This meant that all collars contained some data not associated with livestock movement, but these data points were removed by sorting: firstly, by date and time, and removing all data prior to and after actual collar attachment, and also by defining the geographic limits of the village and removing all data points that did not fall within these geographic boundaries. As the CatLog data file also contains a speed estimate, data points that had recorded excessive speed (>5 m s) were removed.

### 2.4. Time Local Convex Hull

Time local convex hull (T-LoCoH) is a package for analyzing location data developed in the R system for statistical computation and graphics. It is used for constructing home ranges in movement data and explaining spatial–temporal patterns [15]. It generalises the non-parametric utilization construction method and integrates time with space through a scaling that relates distance and time in the construction of local hulls in reference to the velocity of the animal. As a sample for analysis, T-LoCoH produces utilization distributions (UDs) with high fidelity to temporal partitions of space, by taking hulls rather than individual points, and can differentiate various behaviours with internal space. T-LoCoH is also used to construct the UDs from a set of locations by combining minimum convex polygons (MCP) around each point [17]. In each location, time information is used to produce the space used by the animal [18].

### 2.5. Vegetation Sampling

At the study site, 12 transects of 100 m in length were laid at different points to determine the vegetation species composition using a step point method as described by [19]. Twelve transects were determined using Google Earth Pro, where the location of transects was randomly pinpointed within the frequently grazed areas. Along each 100 m transect, a total of 100 individual species were recorded at 1 m intervals [20]. One herbaceous species at or nearest to the pointer was identified and recorded. If the pointer hit the ground, the nearest plant was identified and recorded. The grass species were grouped into Increaser and Decreaser species, depending on their response to grazing [21]. Increaser species are divided into three classes: Increaser I, II and III, and are indicators of poor rangeland. Increaser I is mostly dominant in under-utilized rangeland and in conditions where little or no herbivory takes place; they are generally unpalatable. Grass species that dominate in over-utilized rangelands are referred to as Increaser II, such as sub-climax and pioneer species that produce highly viable seeds and can quickly establish when exposed to the ground. Lastly, Increaser III are generally unpalatable and increase when selective grazing occurs. They are common in over-utilized rangelands [15]. By contrast, Decreaser species are an indicator of good rangeland and dominate in rangelands that are optimally

grazed; however, they decrease in abundance in over- or under-utilized rangelands. These grass species were further classified according to whether they are perennial or annual grasses, depending on their longevity [22]. Perennial grasses can live for more than two years, while annual grasses cannot.

*2.6. MODIS Enhanced Vegetation Index (EVI)*

MODIS EVI was produced from the European Space Agency's Sentinel 2 sensor using a Java script in Google Earth Pro, with an instruction during both wet and dry seasons while the collared animals were being monitored. MODIS EVI minimizes canopy background variations and maintains sensitivity over dense vegetation conditions. The EVI also uses a blue band to remove residual atmosphere contamination caused by smoke and sub-pixel thin cloud clouds. The MODIS EVI product is computed from atmospherically corrected bi-directional surface reflectance that has been masked for water, clouds, heavy aerosols, and cloud shadows. Healthy vegetation will generally absorb most of the visible light (400–700 nm) that falls on it, reflecting a large portion of the near infrared light when there is moisture present in the leaves. On the other hand, unhealthy vegetation reflects light that is more visible and less near infrared light. Bare soils reflect moderately both the infrared and red portions of the electromagnetic spectrum. This study focused on the satellite band that is most sensitive to vegetation information.

*2.7. Data Analysis*

All data were analysed using T-LoCoH software in an R environment following [18,23,24]. T-LoCoH initially facilitates the processing of raw position data into the correct time zone (all raw GPS data are collected using the GMT time zone), followed by sorting into daytime and night-time periods. T-LoCoH allows for the conversion of the geographic projection of a GPS system (latitude, longitude) to a UTM projection. This conversion is necessary to correctly calculate distance and area. Using T-LoCoH, the time stamps on the raw data were corrected to Central African Time. Poor data points (where the positions had <3 satellites for triangulation) were also deleted from the analysis. The animals were penned at night so data were cleaned to include all activities between 6 a.m. and 6 p.m., which were regarded as grazing times. It is however acknowledged that livestock have other activities such as drinking and chewing the cud or walking during daylight hours; however, there was no mechanism on the GPS trackers for discriminating these different activities. Google Earth Pro was used also used to plot livestock distribution movements from the kml files generated by T-LoCoH in the study area.

**3. Results**

*3.1. Wet Season Cattle Movement in the Study Area*

An example of an isopleth representing cattle grazing during the wet season is shown in Figure 2. T-LoCoH provided isopleths that were constructed from hulls, which revealed that the animals spent time grazing on 10% (iso level = 0.1) of all recorded points during the wet season, which is an extremely small area. Different areas represented by small bright red and dark red patches represent areas where the most grazing occurred, forming only 10% and 25%, respectively, of the available grazing area. This is shown by the area in bright red and dark red (Figure 2). Although the animals did move to other grazing areas, they did not spend much time there, shown by edges of the purple color.

*3.2. Wet Season Movement of Sheep in the Study Area*

An isopleth representing a sheep grazing during the wet season is shown in Figure 3. The outputs from the analysis provided the isopleths constructed from hulls. They revealed that most of the grazing occurred in a very small area, forming only 10%, and small portions are also seen in 25% of the available grazing area during the wet season. The greatest elongation (red) around the homestead represents the areas in which most grazing occurred.

The edges (purple color) are an indication of the areas covered by animals, even though no significant grazing occurred in those areas.

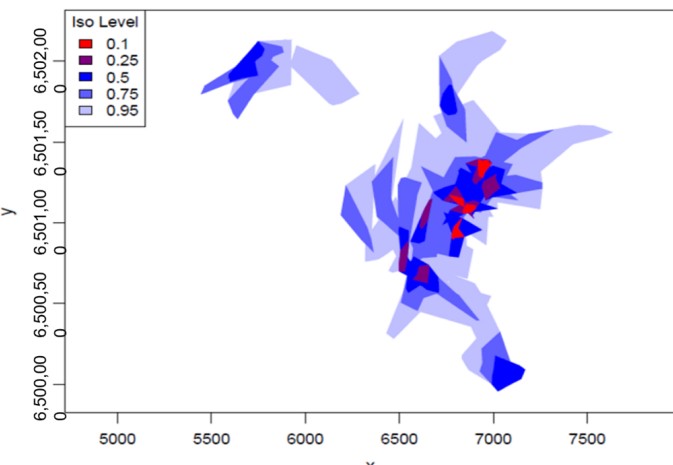

**Figure 2.** An isopleth constructed in different hulls sorted by area, indicating proportions of the total points enclosed by focal cattle during the wet season.

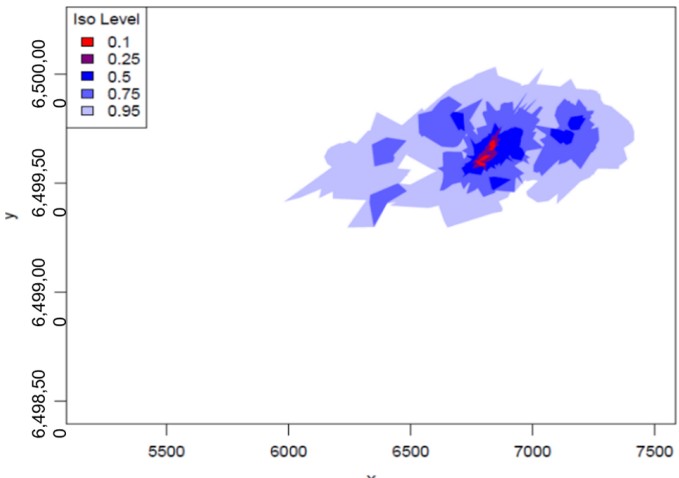

**Figure 3.** Isopleths constructed in different hulls sorted by area, indicating proportions of the total points enclosed by three focal sheep during the wet season.

### 3.3. Dry Season Cattle Movement in the Study Area

The outputs from the T-LoCoH analysis shows an example of representative cattle grazing in the dry season. The results revealed that cattle spent a great deal of their time grazing around homesteads and along the riparian zones (10%) of the grazing land available for livestock grazing (Figure 4). This is shown by the area with greatest elongation (red). The edges of the animal distribution that are shown in purple reveal that the animals roamed around the rangelands even though they did not spend much time grazing.

### 3.4. Dry Season Sheep Movement in the Study Area

An example of representative sheep grazing during the dry season was constructed from hulls sorted by an area during the dry season. The analyses revealed that most of the grazing occurred in only 10%, and small portions in 25%, of the available grazing area around homesteads (Figure 5). The greatest elongation (red) around the homestead represents the areas at which grazing occurred the most. The edges (purple) at which the animal moved around were also revealed for the duration of the study, even though the animals did not spend much time grazing.

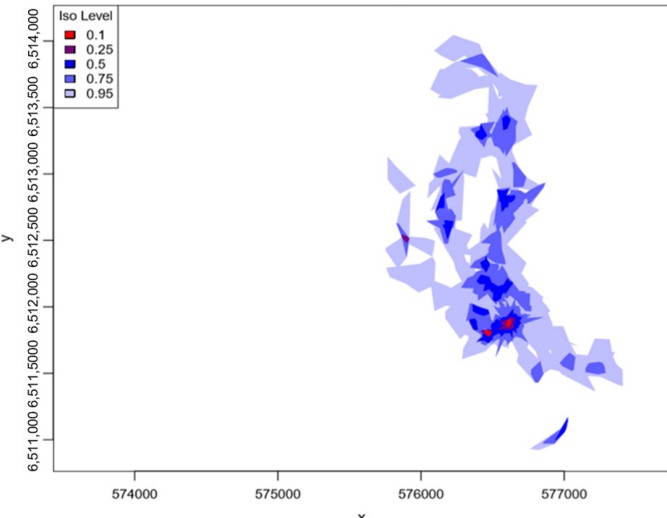

**Figure 4.** Isopleths constructed in different hulls sorted by area, indicating proportions of the total points enclosed by cattle during the dry season.

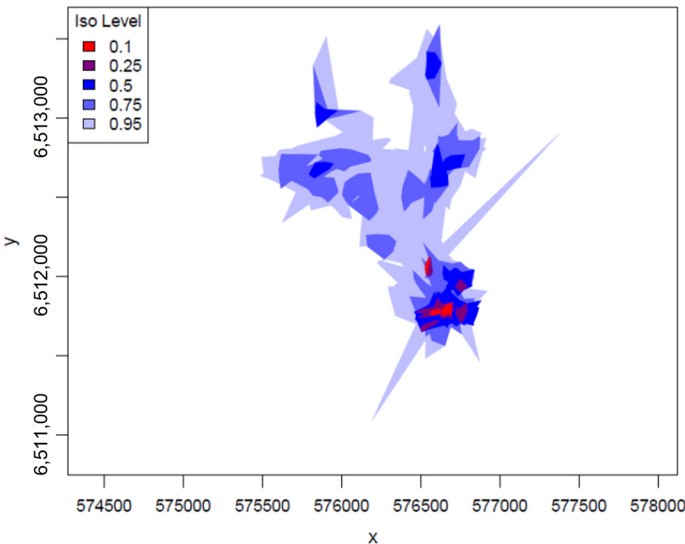

**Figure 5.** Isopleths constructed in different hulls sorted by area, indicating proportions of the total points enclosed by sheep during the dry season.

*3.5. Google Earth Images of Cattle Grazing during the Wet Season*

The Google Earth image (Figure 6) shows the grazing distribution of three cattle during the wet season in the village of Mgwalana. The red, green, and yellow colors represent different animals and show that the animals spent most of their time grazing around the homesteads, around the riparian zones, and a little bit on unimproved grasslands. Grazing occurred during the day as the night-time points were removed from the analysis. However, the animal shown by the yellow color did move and graze on unimproved rangelands during the day in the wet season, while the animal represented by green also grazed in the neighboring village boundary.

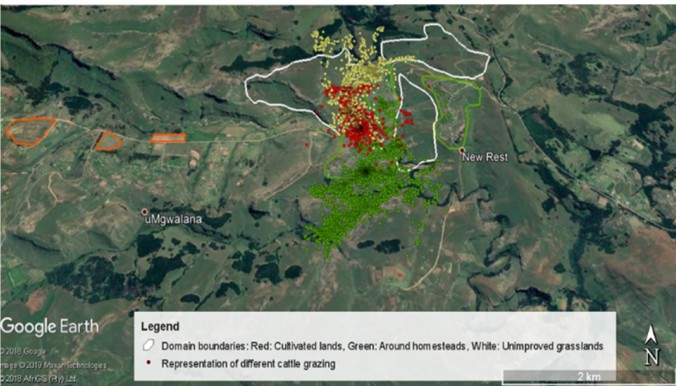

**Figure 6.** Google Earth plot showing the grazing distribution of three different cattle during the wet season.

*3.6. Google Earth Image of Sheep Grazing during the Wet Season*

The Google Earth image for three mature sheep grazing during the wet season shows that the animals spent almost all their time grazing around homesteads (Figure 7). Most of the time was spent in only one domain, with few points on the unimproved grasslands. There were very few points that were recorded along the roads and riparian zones during the whole time of collaring.

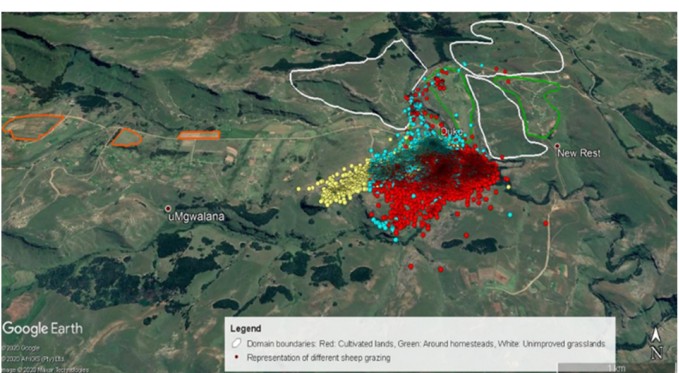

**Figure 7.** Google Earth plot showing the grazing distribution of three different sheep during the wet season.

*3.7. Google Earth Images of Cattle Grazing during the Dry Season*

The Google Earth image shows representative cattle grazing in the dry season in the village of Mgwalana. These animals roamed around looking for grazing, as they were not herded (Figure 8). The image shows that the distribution was concentrated on areas around homesteads, followed by riparian zones and unimproved rangelands. The image shows that the animals are not using the available grazing land effectively as they repeatedly grazed in one area throughout the period of collaring.

*3.8. Google Earth Images of Sheep Grazing during the Dry Season*

Figure 9 shows the grazing distribution of three mature sheep in the village of Mgwalana during the dry season. The image shows that the animals spent most of their time grazing around homesteads, on unimproved lands, and riverbanks to the extent of crossing their grazing border to other areas where they do not belong.

*3.9. EVI of the Areas Were Livestock Grazing Occur*

The enhanced vegetation index from GEE product based on MODIS Terra surface reflectance for the period of livestock collaring was extracted to determine the active green growth of the areas where most grazing (domain) occurred in both the wet and dry seasons

(Figure 10). The mean EVI for production domains from January 2016 to December 2018
were 0.27 for both cultivated lands and unimproved grasslands and 0.29 for areas that were
around homesteads. The EVI for the wet and dry seasons were different, with the wet
season showing high EVI values of 0.3–0.4.

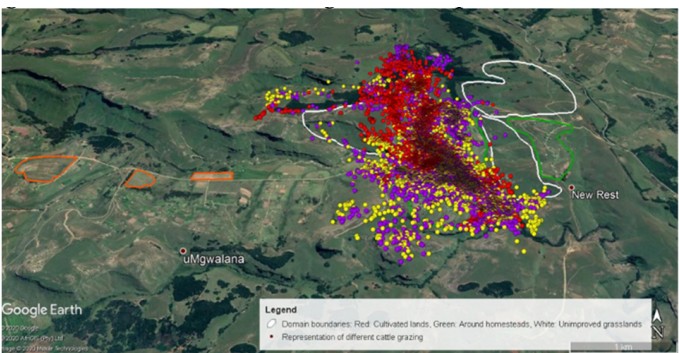

**Figure 8.** Google Earth plot showing the grazing distribution of three different cattle during the
dry season.

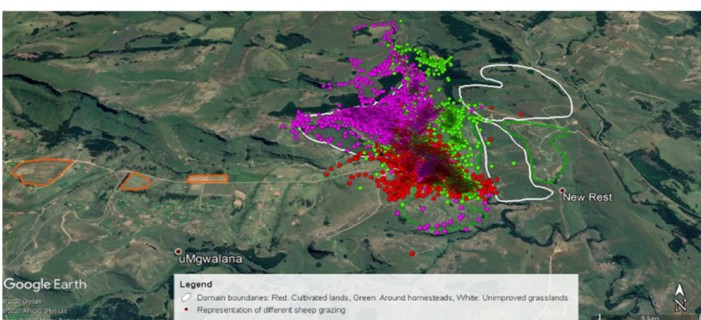

**Figure 9.** Google Earth plot showing the distribution of three different sheep grazing during the
dry season.

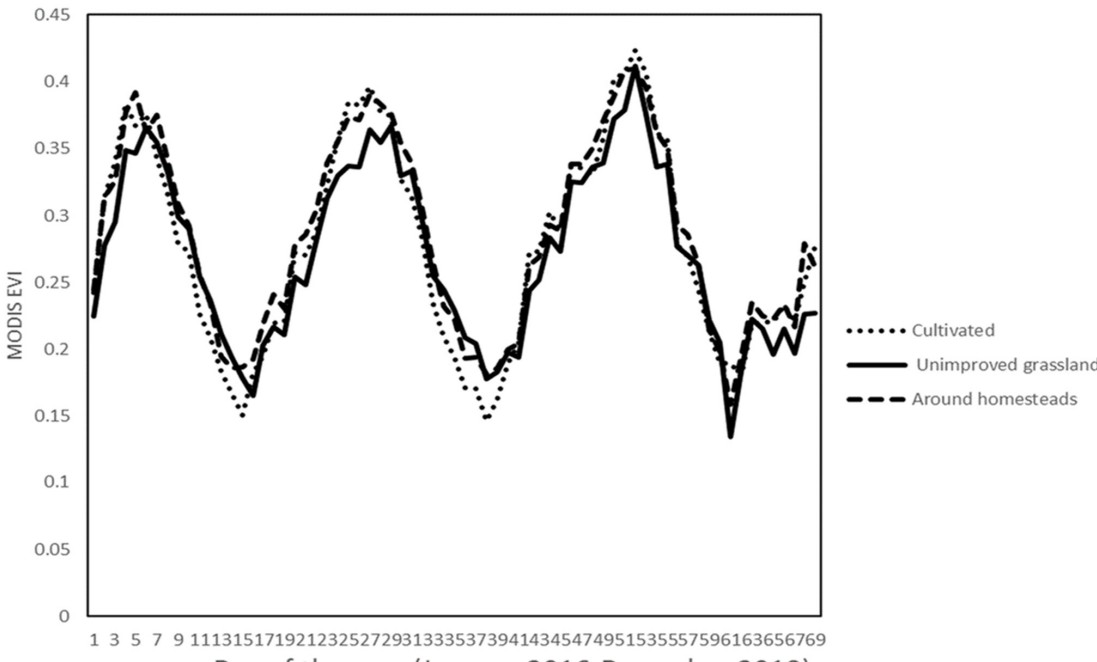

**Figure 10.** MODIS EVI for three grazing domains in the village of Mgwalana for the period of January
2016–December 2018.

*3.10. Distribution of Livestock by Number of Locations and Time*

Table 1 shows the distribution of the grazing livestock by number of locations, steps taken per day, and area covered during grazing in the wet and dry seasons. There is a difference in the number of locations covered by different grazing livestock in both seasons. The total area covered by cattle is much greater than the total area covered by sheep in the wet season, but in the dry season, the total area covered by both grazing livestock does not differ. However, the area covered by sheep during the dry season is greater than the area covered in summer; grazing occurred on a very small proportion of the land (0.05–0.08 ha). Over time, the sampling was not consistent as both cattle and sheep moved location. For both seasons, the number of steps taken per day by both grazing livestock do not differ. The analysis reveals that even though there is a shift in location, it still occurs within a relatively a small area. The average distance (10–50 m) all animals travel in a short time at low speed indicates that the animal spends most of its time during the day grazing. This is the same for all the animals, regardless of gender. However, in the dry season, both cattle and sheep register a higher number of locations than in the wet season. The results show that the distance covered within the location over time occurs in the same area and that the animal spends most of its time during the day grazing.

**Table 1.** Livestock distribution by the number of locations over time during the wet and dry seasons.

| Animal Type | Animal No. Analysed | Total Number of Locations | Av. Number of Steps Travelled | Number of Locations per Day | 10% Area Covered (ha) | Total No. of Points Used | Total Area Covered (ha) | Length of the Recording Period (Days) | Speed (km/hour) |
|---|---|---|---|---|---|---|---|---|---|
| | | | | Wet season | | | | | |
| Cattle | ARC04 | 3911 | 2600 | 90 | 0.02 | 7347 | 126.69 | 79 | 0.70 |
| Cattle | ARC07 | 1741 | 1500 | 93 | 0.01 | 7254 | 397.8 | 78 | 0.74 |
| Cattle | ARC10 | 2900 | 1500 | 94 | 0.01 | 7238 | 122.80 | 77 | 0.85 |
| Sheep | ARC18 | 10176 | 7000 | 93 | 0.18 | 7254 | 48.79 | 78 | 1.35 |
| Sheep | ARC19 | 3189 | 1900 | 90 | 0.09 | 2160 | 39.48 | 24 | 0.07 |
| Sheep | ARC21 | 2052 | 8000 | 69 | 0.18 | 1035 | 59.58 | 15 | 0.87 |
| | | | | Dry season | | | | | |
| Cattle | ARC04W | 8513 | 5200 | 94 | 0.20 | 6016 | 160.50 | 64 | 2.94 |
| Cattle | ARC07W | 8734 | 5200 | 93 | 0.26 | 6720 | 119.71 | 64 | 0.81 |
| Cattle | ARC10W | 6242 | 5800 | 93 | 0.11 | 4185 | 107.39 | 45 | 0.81 |
| Sheep | ARC18W | 8659 | 5300 | 93 | 0.05 | 5952 | 111.54 | 64 | 1.24 |
| Sheep | ARC19W | 8850 | 5300 | 94 | 0.11 | 6016 | 72.52 | 64 | 0.52 |
| Sheep | ARC21W | 8702 | 5200 | 92 | 0.07 | 5888 | 141.03 | 64 | 1.00 |

*3.11. Grass Species Composition in the Grazing Area*

Eight dominant herbaceous species were found in the study site, together with seven grass species and forbs (Table 2). The composition comprises 57% Increaser II species, 26% Increaser I, 17% invader species, and 11% forbs. Of the eight species found in the study area, 50% are of low grazing value, 25% are of moderate grazing value, and another 25% are of high grazing value. All the species found were perennial grasses, except for the forb which was unknown.

**Table 2.** Overall mean percentage abundances of species composition in the village of Mgwalana.

| Plant Species | Ecological Status | Grazing Value | Plant Form | Abundance (%) |
|---|---|---|---|---|
| *Eragrostis plana* | Increaser II | Low | Perennial | 23 |
| *Hyperrenia hirta* | Increaser I | Moderate | Perennial | 26 |
| *Sporobolus africanus* | Increaser II | Low | Perennial | 10 |
| *Forb* | Increaser II | Low | Unknown | 11 |
| *Paspalum dilatatum* | Invader | High | Perennial | 17 |
| *Microchloa cafra* | Increaser II | Moderate | Perennial | 11 |
| *Cynodon dactylon* | Increaser II | High | Perennial | 2 |
| *Cymbopogon plurinodis* | Increaser II | Low | Perennial | 0.25 |
| Total | | | | 100 |

## 4. Discussion

### 4.1. Livestock Grazing Distribution Patterns

The results of livestock grazing distribution patterns for the wet and dry seasons are presented in several ways and all show that during the daytime grazing periods, livestock spend a great deal of their time close to human features. The preparation of hull sets presented for cattle and sheep using T-LoCoH shows that in both seasons, animals spend most of their time grazing on 10% and 25%, a very small area (1–5 ha) near anthropogenic features, for example, homesteads, along roadsides, in abandoned cultivated lands, and riparian zones. This strong association is possibly due to the active green growth around these areas; even though the grazing resource is very limited, the grass is generally very short, and is mainly stoloniferous species such as couch grass (*Cynodon dactylon*) and kikuyu (*Pennisetum clandestinum*) and does not offer much bulk forage [10]. However, livestock are able to 'learn the landscape' [25] and know that they have to come home, which may be one of the reasons for their strong association with anthropogenic features. While these areas provide key resources for livestock, the foraging behavior of sheep can be explained by their grazing mechanism, as they are adapted to selective grazing and allometry of their food intake, which restricts them to feeding on tall grasses and herbs. Stated by [26], grasses that grow among human features (homesteads in this case) are ideally suited for sheep's small bites, and this is also supported by [27]. On the other hand, cattle grazing pattern for both seasons were similar, with a grazing preference around homesteads. This suggests that in both seasons, forage on unimproved land is not being used effectively as it might be. The sum of these findings provides an important insight into livestock foraging preference on areas that are close to homesteads in the study site. Both sheep and cattle have demonstrated a strong preference for key resources that are available in these areas, whereas grass in the rangelands is mostly avoided. However, these are preferred in the dry season by cattle when preferred vegetation in key areas have become depleted.

Furthermore, photosynthetically active green growing grasses generally indicate high nutrient-enriched sites and are adjacent to stockades and houses, riparian zones, and soil conservation structures [10]. These represent the classic grazing lawns of the natural African rangelands [27], on which most grazing occurs. The cleared surfaces associated with human habitation (roads, footpaths, cultivated areas) create larger runoff areas that provide water to small patches (run-on areas). These patches, akin to the patch dynamics described by [28], provide short, green, nutrient-rich grass throughout the year. Continued revisiting of these sites replenishes nitrogen and other nutrients on a daily basis through dunging and urination. According to [29], livestock avoid grazing near faeces, but when decomposed and highly nutritious biomass is available later, they will prefer such grazing, resulting in limited grass growth at a later stage. This daily enrichment is similar in nature to the effect of resident indigenous antelope that create and occupy grazing lawns (bontebok, blesbok, and black wildebeest). Although there is limited understanding about

the sustainability and dynamic of these lawns, the authors in [30] show that "periods of positive plant growth following the onset of rains coincide with periods of low N turnover rates, whereas higher rates occur late in the wet season following plant senescence and throughout the dry season". These higher rates may be what attract herbivores to stay on these sites, even when alternative grazing in the natural grasslands may be of better quality.

More extensive free-range grazing on upland grasslands did occur, but the density of animals was very light. This may be associated with the small livestock numbers which, according to [31], tend to function as a single unit. Additionally, the authors in [32] found that small herds usually graze close to one another and those in [31] argue that cattle are social animals that tend to associate in groups when grazing and normally behave similarly, moving to watering points, grazing, and resting together. The analysis of the information provided on location changes over time; the distance travelled and the speed at which both sheep and cattle moved shows that they spend most time during the day grazing. According to [33], the speed for resting animals should be zero, high for a moving animal, and low for a grazing animal. In this study, cattle travelled in search of grazing because they were not monitored, which is different from a study by [1], which found that cattle that were monitored travelled less distance in the wet season, so saving energy, which resulted in weight gains. However, both cattle and sheep travelled long distances in search for food in the dry season. Furthermore, the authors in [34] state that the number of locations covered by the animal looking for grazing during the dry season contributes to weight loss. However, the authors in [12] argue that animals decide where to graze based on their perception and knowledge of consumed plants in the area and their potential choice memory, which later develops a map-like representation of the different areas within the rangeland. Animals have a long-term memory [12] which makes them return repeatedly to areas where they previously grazed, in search of forage until that forage is depleted.

### 4.2. Factors Affecting Grazing Distribution

The grazing distribution across the landscape suggests that there are several factors that affect how livestock use the landscape, as some of the grazing land was not used by free-ranging animals. The factors observed in this study include topography, distance from water, and plant composition and these were also found by [12]. The grazing patterns, according to [8], may reduce stream bank stability, reduce vegetation cover, and increase soil erosion. Areas located along the steep slope received less grazing in this study, possibly because the steep slope was far from the watering point and there was no available shade for rest, with shade being one of the factors affecting livestock distribution [8]. It is posited that during the dry season, animals move further in search of grazing if the resource is limited. However, this study reveals the opposite. Possibly, the animals do not move far in search of forage in order to conserve energy. The patterns of animal distribution in this study show that unrestrained domestic livestock will only move to natural grasslands areas (usually at higher elevation) for short periods during the wet and dry season grazing cycle. Additionally, the authors in [12] state that topography is one of the causes of poor grazing distribution, because cattle prefer flat and gently rolling terrain, such as valley bottoms, level benches, and low areas between drainage. The fact that cattle were seen grazing on a steep slope during collaring suggests that they may be unwilling rather than unable to graze on a steeper slope. However, sheep which are smaller, surefooted, and more agile can make more use of steeper slopes but are limited to where rugged terrain can limit their landscape use [12]. The collared animals demonstrated that livestock prefer certain grazing sites to others because of the terrain. In addition, roads play a role as the study reveals that animals grazed along the road, which according to [8], enables the animals to travel and graze on the vegetation alongside.

Distance to water (the river, in this case) may be a strong motivator of grazing distribution for both cattle and sheep. Water acts as a limitation at which the mechanism for foraging operates. According to [12], livestock need free access to water for improved production. Distant water sources decrease livestock production efficiency because they

use energy to travel from the grazing area to watering points, thus making availability a major source of poor grazing distribution across the rangeland. However, plants that are near the watering points are heavily grazed, which results in reduced forage production. In another study where animals were fitted with GPS collars, the authors in [12] found that watering points are the main factors determining the distribution, movement, and concentration of grazing animals. Their results reveal that areas that are close to watering points receive more grazing than areas that are far from the watering points. Lastly, different livestock species have different forage preferences, strongly influencing grazing distribution. Cattle prefer grazing on grasses and tend to avoid bushes; however, sheep prefer short green grass. Species composition also provides an insight into the major role that livestock grazing distribution has on the rangeland [35]. Although different plants may be found in the rangeland, they receive different grazing pressure because of their chemical composition [32]. In this study, the most abundant species in the area was *H. hirta*, a robust perennial C4 grass moderately resistant to continuous grazing, implying that the rangeland is disturbed and overgrazed, as it grows along the road verge and in disturbed soils or abandoned arable lands. It is also known for poor nutritional quality, which may be mostly avoided by livestock unless there is no alternative [26]. Another species that was abundant in the study area was *E. plana*, which grows in disturbed and overgrazed soils. Although this species tolerates overgrazing because of its strong root production, succession ability, drought tolerance, and its ability to compete for resources [20], it has low forage value and can result in livestock economic losses.

### 4.3. Livestock Grazing Distribution and Implications for Proper Grazing Management

From a grazing management perspective, it is important to understand the impacts of livestock grazing on species composition as they form part of key resources that livestock can exploit. The extent to which the grazing distribution by livestock or livestock owner occurs is also important as the use of key areas for grazing is often driven by the decision made by livestock or the herder. In addition, the authors in [22] argue that proper grazing management can improve the grazing distribution of livestock. In this study, animals were not herded but they were left on their own to select their preferred grazing sites. Herding can be more effective and as it is a proven tool for controlling livestock distribution, especially when livestock grazing behaviour is considered. In the over-utilized areas of the rangelands, improved grazing distribution could result in an increased stocking rate because of the more available forage in the rangeland [6]. According to [22], the herder should remain with the animals in the rangelands until they get used to the area. Moreover, the authors in [8] state that livestock herding, where people move livestock on horseback from one location to another has long been recommended as a management tool to modify grazing patterns of cattle for the even utilization of grasses. Although livestock herding is costly and requires labour, while livestock owners in communal rangelands rely on social grants, the over-utilization of preferred areas such as riparian zones and around homesteads makes herding necessary to reduce the negative impacts caused by heavy grazing.

### 5. Conclusions

The analysis of livestock grazing distribution on the landscape reveals that animals spend most time grazing around homesteads, along roads, and along riparian zones, in both the wet and dry seasons. The possible explanation for this grazing habit in both sheep and cattle is linked to the repeated revisiting of areas that are actively green for most of the year. An abundance of *E. plana* and *C. dactylon* between the homesteads and on the run-on areas near roadways was also observed, which animals favored over other grasses in the rangelands. The use of key resource areas by both sheep and cattle results in exploitation of resources and may later contribute to accelerated degradation of these resources. Livestock owners or herders need to observe where their animals spend most time grazing, as the bulk of the rangeland is not used. Extending grazing areas will improve grazing distribution, improve species composition, and forage quality

and quantity. It is important to inform owners or herders about the implications of the current grazing distribution for the rangeland's condition, and for households to have herders who will move livestock into areas that are lightly grazed for a more even grazing distribution. Furthermore, findings from this study have an important implication for rangeland management policy in communal areas, such as the introduction of a committee set up by the local people to examine the management of rangeland use.

This can be a significant intervention as the committee would ensure that the areas are divided, rested, and grazed at times when the growing season has lapsed. Encouraging livestock owners to assign some of the grazing land during the growing season will also improve species composition and cover, while reducing degradation. This is because the unmonitored grazing distribution of livestock, especially during the wet season, has negative impacts on species composition. Therefore, this study suggests a need to monitor livestock movements in communal rangelands. Alternatively, introducing livestock herding to livestock owning households is a possibility, which could improve rangeland species composition and production. Livestock herding is a process whereby people move livestock in a rangeland from one place to another for an even rangeland. A herder in communal rangelands could move livestock to different grazing areas for improved, even grazing distribution, as the herder purposely relocates animals to alternative sites without harassing them from their preferred sites; harassment often makes them return to their preferred site. This initiative could be incentivized by the government and included in policies that seek to improve rural development as it will also improve the environmental state of these rangelands and people could also benefit from the rangeland's goods and services that improve their livelihoods. Some of the incentives might include capacity development in animal husbandry and rangeland management.

**Author Contributions:** Conceptualization, B.G. and A.R.P.; methodology, B.G. and A.R.P.; software, B.G., A.R.P. and T.C.Z.; validation, B.G., A.R.P. and T.C.Z.; formal analysis, B.G., A.R.P. and T.C.Z.; investigation, B.G.; resources, A.R.P.; data curation, T.C.Z.; writing—original draft preparation, B.G.; writing—review and editing, A.R.P.; visualization, B.G.; supervision, A.R.P.; project administration, B.G. and A.R.P.; funding acquisition, A.R.P. All authors have read and agreed to the published version of the manuscript.

**Funding:** This research was funded by the Agricultural Research Council (Professional Development Programme), Water Research Commission (K5/2400/4) and National Research Foundation.

**Data Availability Statement:** Data for this study is kept for privacy issues as it includes detailed location of where livestock were grazing.

**Acknowledgments:** The Authors would like to acknowledge the communal farmers in Mgwalana village for their assistance during data collection and animal handling.

**Conflicts of Interest:** The authors declare no conflict of interest.

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
