# Peer review of "Assessing Livestock Grazing Distribution in Communal Rangelands of the Eastern Cape, South Africa: Towards Monitoring Livestock Movements in Rangelands"

_land, doi:10.3390/land12040760_

Round 1

Reviewer 1 Report

Introduction

There appear to be formatting issues with spacing between words throughout the manuscript.

L29 suggest comma after “degraded”

L36 “South Africa”?

L56 italicize genus and species names? Throughout the manuscript.

L58 “affects”

L59 “…of a patchy…”

Materials and Methods

L88 “…a few goats…”

L99 “warm cool” ?

L107 sheep breeds? Genus/species.

L114-116 this description is confusing

L160 is this 12, 100 m transects? Or is each transect 12x100 m? L162 infers the former, but it would be good to clean up this description.

L204-206 were the cattle penned at night? Why apply grazing uniformly during daylight? I understand that your GPS trackers did not allow for this but this is an important point, you really need to be able to distinguish these behaviors.

Results

L214 “… grazing on 10% …)

L225 This sentence seems incomplete.

L227 suggest “… and small portions in 25% of the…”

L237 suggest “…shows representative cattle grazing in the dry season.”

L247 “…example of representative…”

L261 “small portion”

L262 was grazing actually restricted to the daylight hours or were these points just removed from analysis?

L347 Why do they “have to come home”? are they penned and fed at night?

Discussion

L414  again, are they being fed in and around the homesteads?

L454 “(Scheffer-Basso et al. 2016)”

L476 “return to”

L481 is it “too costly”?

L482 “ such as”

Conclusions and recommendations

L485 delete “in”

Overall, nice paper on an important topic. Generally well written but needs a few editorial items cleaned up as indicated. There also needs to be a  more detailed description of animal management, if the animals are being penned at night and fed for instance, this will affect your results and should be discussed. On a related note, it is also not completely clear why night grazing was excluded and the discussion of grazing behavior/location would be stronger if you could further justify why you allocated the entire day to grazing. There is literature and probably observational data that would inform an allocation to resting, ruminating, etc…      

Reviewer 2 Report

Dear Authors,

the article entitled “Assessing livestock grazing distribution in communal rangelands of the Eastern Cape South Africa: towards monitoring livestock movements in rangelands” seems interesting but requires some modifications on your part. Apart from some issues regarding editing, for which I suggest following the guidelines of the journal, the different parts of the article seem excessively fragmented and could be better amalgamated. Below you will find some general suggestions and others specific to the individual parts, I hope they can help you.

In addition to the affiliation, it is necessary to indicate the email address of all the authors and not only of the contact person.

Please organize the bibliographic references according to the journal guidelines: in the text they must be numbered (in order of appearance) in square brackets, in the "references" they must be numbered (in order of appearance).

Please check for spaces throughout the document.

For all figures: check the font, line spacing, etc. in the title, following the guidelines of the journal.

Try to reduce the number of subparagraphs, many of them can be grouped, in this way the speech becomes more linear and understandable.

Abstract: insert one or two initial sentences to introduce the topic of analysis and better specify the objective of the work.

Introduction: it lacks how the article is structured and perhaps better specify the objective of the work.

Materials and methods:

·         line 127 what is (Pers. Comm.)? I think can be deleted.

·         line 181_182: perhaps it is better to remove the acronym from the title and then insert it in the text first in full and then the acronym.

·         Perhaps it is better to limit the description relating to the IT part, it seems to me that it creates a bit of confusion.

Results:

·         Check for spaces, for example “10 %” must be “10%”.

·         line 225: what does mean “The representative sheep grazing during the wet season”? please explain better this sentence.

·         As regards the comments on the tables/figures, try to follow the order of the same and do not insert comments that are too distant.

·         I would suggest combining the comments for wet season and dry season, and the figures could also be placed next to each other to show the differences in the movements of the different animals.

·         table 1: the title is missing; the second part provides the data relating to the "dry season" while what does the data in the first part indicate? “Wet season? the indication must be placed after the title of the columns, not before. Some column titles are in bold some are not, please review the editing.

·         line 334: there is another table 1, please revise table numbering.

Discussions: you could simplify and unify the paragraphs a bit, some of the discussions could be used to broaden the conclusions.

Conclusions: the title should only be “Conclusions”. Expand this part a little, perhaps also adding something about future developments of the work.

The whole part relating to the attribution of the parts, funds, etc. is missing.

Round 2

Reviewer 2 Report

Dear Authors,

thank you for the effort you have made to improve the article in such a short time. It seems to me that you have responded precisely to all my requests and followed my suggestions. I ask you just a little further effort because I still have something to point out:

1.       “Please organize the bibliographic references according to the journal guidelines: in the text they must be numbered (in order of appearance) in square brackets, in the "references" they must be numbered (in order of appearance)”. In relation to this point, I have seen that you have put the bibliographic references with numbers and in square brackets, but the order is still wrong. Perhaps my suggestion is clearer with an example: if Odadi et al is the first author you cite, it must not be [23] but [1] and in the bibliographic references at the end of the text it must appear as first in the list. The order is by citation and not alphabetical.

2.       Thanks for following the suggestion to show the figures together so that it would be easier to compare. Pay attention to the text of the title, for example for figure 2 and figure 3, in the first case "sorted by area" is between commas, in the second case not, try to homogenize. Then it applies to all, try to make invisible the edges of the box where you wrote the title.

3.       “The whole part relating to the attribution of the parts, funds, etc. is missing”. Even if you added the information about the Acknowledgments but the information about Author Contributions, Funding, Data Availability Statement, Conflicts of Interest etc. are still missing, please provide them.

Good luck.
